# Compact Ga_2_O_3_ Thin Films Deposited by Plasma Enhanced Atomic Layer Deposition at Low Temperature

**DOI:** 10.3390/nano12091510

**Published:** 2022-04-29

**Authors:** Yue Yang, Xiao-Ying Zhang, Chen Wang, Fang-Bin Ren, Run-Feng Zhu, Chia-Hsun Hsu, Wan-Yu Wu, Dong-Sing Wuu, Peng Gao, Yu-Jiao Ruan, Shui-Yang Lien, Wen-Zhang Zhu

**Affiliations:** 1School of Opto-Electronic and Communication Engineering, Xiamen University of Technology, Xiamen 361024, China; 1922031005@stu.xmut.edu.cn (Y.Y.); xyzhang@xmut.edu.cn (X.-Y.Z.); chenwang@xmut.edu.cn (C.W.); 2022031161@s.xmut.edu.cn (F.-B.R.); 2022031203@stu.xmut.edu.cn (R.-F.Z.); chhsu@xmut.edu.cn (C.-H.H.); wzzhu@xmut.edu.cn (W.-Z.Z.); 2Fujian Key Laboratory of Optoelectronic Technology and Devices, Xiamen University of Technology, Xiamen 361024, China; 3Department of Materials Science and Engineering, Da-Yeh University, Dacun, Changhua 51591, Taiwan; wywu@mail.dyu.edu.tw; 4Department of Applied Materials and Optoelectronic Engineering, National Chi Nan University, Nantou 54561, Taiwan; president@ncnu.edu.tw; 5Fujian Provincial Key Laboratory of Nanomaterials, Fujian Institute of Research on the Structure of Matter, Chinese Academy of Sciences, Fuzhou 350002, China; peng.gao@fjirsm.ac.cn; 6National Measurement and Testing Center for Flat Panel Display Industry, Xiamen Institute of Measurement and Testing, Xiamen 361004, China; ryj@xmjly.com

**Keywords:** Ga_2_O_3_ thin film, substrate temperature, atomic layer deposition

## Abstract

Amorphous Gallium oxide (Ga_2_O_3_) thin films were grown by plasma-enhanced atomic layer deposition using O_2_ plasma as reactant and trimethylgallium as a gallium source. The growth rate of the Ga_2_O_3_ films was about 0.6 Å/cycle and was acquired at a temperature ranging from 80 to 250 °C. The investigation of transmittance and the adsorption edge of Ga_2_O_3_ films prepared on sapphire substrates showed that the band gap energy gradually decreases from 5.04 to 4.76 eV with the increasing temperature. X-ray photoelectron spectroscopy (XPS) analysis indicated that all the Ga_2_O_3_ thin films showed a good stoichiometric ratio, and the atomic ratio of Ga/O was close to 0.7. According to XPS analysis, the proportion of Ga^3+^ and lattice oxygen increases with the increase in temperature resulting in denser films. By analyzing the film density from X-ray reflectivity and by a refractive index curve, it was found that the higher temperature, the denser the film. Atomic force microscopic analysis showed that the surface roughness values increased from 0.091 to 0.187 nm with the increasing substrate temperature. X-ray diffraction and transmission electron microscopy investigation showed that Ga_2_O_3_ films grown at temperatures from 80 to 200 °C were amorphous, and the Ga_2_O_3_ film grown at 250 °C was slightly crystalline with some nanocrystalline structures.

## 1. Introduction

Gallium oxide (Ga_2_O_3_) thin film has attracted wide attention owing to its wide band gap, high breakdown electric field, and high optical transparency. It has been applied to some high-performance devices, such as deep ultraviolet light detectors [1,2], photodiodes [3,4], and transparent field effect tubes [5,6]. Ga_2_O_3_ is a kind of oxide semiconductor material; it has a variety of crystal phases, including α, β, γ, δ, ε. The most stable and widely studied phase is the β-Ga_2_O_3_ phase. The other metastable phases can be transformed into relatively stable β-Ga_2_O_3_ after high-temperature thermal annealing. There are many methods to prepare Ga_2_O_3_ thin film, such as radio frequency (RF) [7,8] magnetron sputtering, pulsed laser deposition (PLD) [9,10], molecular beam epitaxy (MBE) [11,12], metal-organic chemical vapor deposition (MOCVD) [13,14], and atomic layer deposition (ALD) [15,16,17]. Although magnetron sputtering and PLD can prepare Ga_2_O_3_ thin films on a relatively low-temperature substrate, the most obvious shortcoming is that the film surface cannot show a large area of uniformity, and the crystal quality needs further improvement. Ga_2_O_3_ films prepared by MBE and MOCVD need to be carried out at a relatively high substrate temperature, such as 700–1000 °C. Compared with other deposition methods, ALD is considered to be one of the most advantageous deposition methods because of its good thickness control and uniform film growth. In addition, compared with traditional ALD, plasma-enhanced atomic layer deposition (PEALD) can improve the crystallinity of the film and enhance the effect of plasma in the whole ALD reaction process. It can have better control of the film’s growth in lower temperature environments and reduces the heat load in the reaction. Therefore, PEALD is widely used in film growth. Studying the properties of Ga_2_O_3_ films prepared by PEALD at a low temperature is a popular topic. For example, Dong-Won Choi et al. deposited Ga_2_O_3_ film by ALD at a low deposition temperature (150–250 °C) using gallium tri-isopropoxide (GTIP) as the gallium source and H_2_O as the oxygen source. The growth rate of Ga_2_O_3_ film increased sharply and saturated to 0.25 nm/cycle [18]. Using trimethylgallium (tmga) and ozone (O_3_), Daniel Hiller et al. observed that Ga_2_O_3_ growth per cycle (GPC) was ~0.4 Å/cycle and ~0.5 Å/cycle in thermal ALD at 250 °C and 300 °C. The GPC of Ga_2_O_3_ reaches ~0.7 Å/cycle in PEALD at 75 °C [17]. It can be seen that PEALD can obtain a higher deposition rate using a lower deposition temperature than that of thermal ALD. There also have been many reports about the selection of precursors. Saidjafarzoda Ilhom et al. prepared Ga_2_O_3_ films using trethylgallium (TEG) and Ar/O_2_ as the precursor system. The substrate temperature range was 150–240 °C, and the RF plasma range was 30–300 W. The GPC of the film ranged from 0.69 to 1.31 Å/cycle [19]. The precursors used by Ramachandran et al. were Gallium tetramethylheptanedionate (Ga(tmhd)_3_) and oxygen. They prepared the films in the temperature range of 100–400 °C and an RF power of 300 W. Under these conditions, the film growth rate was 0.1 Å/cycle [20].

In this study, Ga_2_O_3_ films were prepared using tmga and oxygen as precursors through PEALD at a plasma power of 2500 W. The substrate temperature varied from 80 to 250 °C. The growth temperature effect on optical properties, surface morphology, chemical element distribution, and structural properties of the deposited PEALD Ga_2_O_3_ film were comprehensively studied. The effects of different substrate temperatures on the deposition mechanism and properties of Ga_2_O_3_ thin films were discussed to obtain high-density Ga_2_O_3_ thin films with few defects.

## 2. Materials and Methods

In this work, sapphire and silicon were used as deposition substrates. The sapphire substrate was cleaned with deionized water, ethanol, isopropyl alcohol, and deionized water for 15 min in sequence, respectively. The sapphire substrates were then blown by N_2_ and transferred to a vacuum drying cabinet for more than 20 min in order to remove water vapor. For the Si substrates, they were cleaned through a standard Radio Corporation of American (RCA) cleaning process and then soaked in 2% hydrogen fluoride solution for 1 min to remove the surface natural oxide layer. They were then cleaned in deionized water and finally dried with an N_2_ gas gun. After cleaning, the substrates were transferred into the ALD reaction chamber. Ga_2_O_3_ films were deposited on different substrates (Si and sapphire) at 80, 100, 150, 200, and 250 °C using tmga and oxygen plasma at 2500 W in a commercial PEALD system (Picosun R-200, Espoo, Finland). The plasmas were produced in a microwave cavity using inductive coupling of RF power (Litmas RPS, Advanced Energy, Denver, CO, USA) with the mixture of Ar and O_2_ gases, which belonged to a high-density remote plasma system. According to our previous experimental study [21,22], 2500 W can form high-density oxygen free radicals, which enables it to carry out a good oxidation saturation reaction and avoid plasma bombardment on the surface of the film. Therefore, high-density and high-quality gallium oxide films can be obtained at the power of 2500 W. N_2_ gas was used as the carrier gas with the flow rates of 120 sccm for tmga. Tmga was stored at 0 °C in stainless bottles. The deposition process of PEALD Ga_2_O_3_ sequentially included: tmga pulse time (0.2 s)-N_2_ purge time (4 s)-O_2_ plasma processing (28 s)-N_2_ purge time (4 s). During the deposition, the total number of cycles for all the samples is 600. Table 1 is a list of the growth parameters of the PEALD Ga_2_O_3_ films.

The thickness and refractive index of the Ga_2_O_3_ films deposited on silicon wafers were characterized by a spectroscopic ellipsometer (SE, SENTECH SE 800 DUV, Berlin, Germany). The refractive index of the Ga_2_O_3_ thin films was obtained from the SE characterization assuming a Tauc–Lorentz model. The optical transmittance of the films deposited on the sapphire substrates was obtained by a UV-vis spectrophotometer (Lambda850, PerkinElmer, Waltham, MA, USA) at a wavelength between 200 and 800 nm at room temperature. The structural properties of the Ga_2_O_3_ films were examined by conventional θ-2θ X-ray diffraction (XRD, Rigaku TTRAXIII, Ibaraki, Japan) using a copper Kα emission line. The chemical compositions and bonding states were characterized by X-ray photoelectron spectroscopy (XPS, ESCALAB 250Xi, Thermo Fisher, Waltham, MA, USA). The size of the Ga_2_O_3_ films for XPS measurement is 5 × 5 mm^2^. Ga_2_O_3_ films were fixed on the XPS test bench using conducting resin. Non-test surfaces of the Ga_2_O_3_ films were marked and distinguished from test ones. The total acquisition time was 136.1 s. XPS measurements were recorded using monochromatized Al K Alpha as the excitation source with a spot size of 400 µm^2^. According to the reported literature [23], the chemical state of Ga could not be changed after Ar^+^ pre-sputtering on the sample’s surface. Nevertheless, a few other reports [24,25,26] inferred that the introduced Ar^+^ bombardment had a crucial effect on the oxidation states. Hence, in order to eliminate the influence of Ar^+^ pre-etch on the formation of lower Ga oxidation valence states, the XPS measurements were executed without Ar^+^ pre-sputtering. The thickness of the Ga_2_O_3_ thin films was also determined by X-ray reflectivity (XRR, Rigaku TTRAXIII, Ibaraki, Japan) and HR-S/TEM (FEI Talos F200X, Hillsboro, OR, USA). XRR is a nondestructive technique frequently used for measuring the thickness, density, and roughness of thin films [27]. The crystal structure and interfacial layers of the Ga_2_O_3_ thin films were investigated by HR-S/TEM. Energy dispersive X-ray (EDX) analysis for the elemental mapping of the Ga_2_O_3_ thin films was performed with the same instrument.

## 3. Results and Discussion

Figure 1 shows the changes in the film’s GPC with a substrate temperature from 80 °C to 250 °C. The GPC is defined as the thickness divided by the number of cycles. The thickness of the Ga_2_O_3_ films prepared at the 600 cycles with a substrate temperature varying from 80 °C to 250 °C is 40.86, 39.90, 38.65, 37.87, and 37.81 nm, respectively. It can be seen that the change of thickness is not obvious, but the GPC of the film gradually decreases and tends to saturate with the increase in substrate temperature to 200 °C. When the temperature is 80 °C, the film is the thickest, corresponding to the largest deposition rate of 0.068 nm/cycle. When the growth temperature increases to 200 °C, the GPC decreases to 0.063 nm/cycle, as shown in Figure 1. Therefore, the substrate temperature has little effect on the deposition rate, which ranges in GPC from 0.068 nm/cycle to 0.063 nm/cycle when the substrate temperature is 80–250 °C. This phenomenon may be related to the physical and chemical adsorption of the precursor. Some precursors have a low activity due to a low temperature. The higher growth rate observed below 100 °C may be due to precursor condensation onto the substrate or the incomplete removal of the reaction by-products at low temperatures, giving rise to some organic residues from the precursor ligands incorporated into the film [20]. On the other hand, it is caused by chemisorption. In this study, the precursor chemical reaction can be elaborated by the following equations:Si-OH + Ga(CH_3_)_3_ → Si-O-Ga(CH_3_)_2_ + CH_4_(1)
Si-O-Ga(CH_3_)_2_ + O* → Si-O-Ga-OH + CO_2_ + H_2_O(2)

It is known that the precursor of Ga(CH_3_)_3_ reacts with a hydroxyl group on the surface of a silicon wafer to decompose into Si-O-Ga(CH_3_)_2_ and CH_4_ gas in the whole ALD reaction system. When O_2_ plasma participates in the second semi-reaction, Ga_2_O_3_ is generated. Therefore, the precursor may condense because of the low deposition temperature resulting in a high deposition rate. As the substrate temperature rises, the desorption ability between the precursor and the deposition surface will be enhanced, which will cause a decrease in the deposition rate.

Figure 2a shows the XRD patterns of the Ga_2_O_3_ films prepared at various substrate temperatures. No obvious peaks were detected in the deposited Ga_2_O_3_ film at the temperature of 80–200 °C, indicating that the Ga_2_O_3_ films are uncrystallized or contain some localized microstructure, such as nano-crystals. The amorphous characteristics of the films can be attributed to the low growth temperature [8,28]. At the same time, it is found that there is slight crystallization in the film at 250 °C. By comparing PDF#06-0503, it can be seen that there is a broad peak at 55.114°, corresponding to the (116) crystallization plane of the α-Ga_2_O_3_ film. The density of Ga_2_O_3_ films was measured by XRR. Figure 2b shows the XRR pattern of β-Ga_2_O_3_ films. The density of the Ga_2_O_3_ thin films as a function of growth temperature is shown in Figure 2c. The film density increases with the increase in the growth temperature. It is noteworthy that the density decreased to 5.154 g/cm^3^ at 80 °C due to insufficient surface reaction. With the continuous increment of deposition temperature, the density gradually increased. The highest density of 5.325 g/cm^3^ was achieved at 250 °C, which was slightly lower than the β-Ga_2_O_3_ bulk density of 5.95 g/cm^3^ [29,30]. As demonstrated below, the ALD deposited on the Ga_2_O_3_ films were amorphous, leading to a lower density [8]. Compared with the previously reported Ga_2_O_3_ density of 5.95 g/cm^3^ prepared by edge-defined film-fed growth (EFG) [30], the density of Ga_2_O_3_ film is lower, which is on account of the non-crystalline structure of Ga_2_O_3_ film prepared by ALD process [28]. The density of Ga_2_O_3_ prepared by Z. Yu et al. using MBE was 5.3 g/cm^3^ [31]. The density of Ga_2_O_3_ prepared by M. Passlack et al. using the electron—beam specific fuel was 5.15 g/cm^3^ [32]. Nieminen et al. prepared a Ga_2_O_3_ film by atomic layer epitaxy, with Ga(acac)_3_ and ozone as the precursor system. They found that the density of the film is 5.6 g/cm^3^ [33]. The density of β-Ga_2_O_3_ measured by Richard et al. was 5.12 g/cm^3^ [34]. It can be seen that the density of Ga_2_O_3_ film grown in this study is within the theoretical range and superior to ordinary deposition techniques. In Table 2, a comparison has been made for the properties of Ga_2_O_3_ films prepared by different methods. Compared with other methods, the Ga_2_O_3_ films deposited by PEALD have a higher film density, which is also slightly better than the results in Ref. [34]. Moreover, the roughness of the film in Ref. [34] is 0.31–0.44 nm, while the roughness in our study is only 0.187 nm. Therefore, the gallium oxide film prepared in this study had a relatively high density with a very smooth surface.

The chemical valence state and composition of Ga_2_O_3_ thin films were analyzed by XPS measurements. Figure 3a shows the investigation spectra of the deposited Ga_2_O_3_ films at different substrate temperatures. The spectra were mainly marked with peaks associated with Ga, such as Ga 3s, Ga 3p, Ga 3d, Ga 2p_1/2_, and Ga 2p_3/2_, as well as peaks associated with O, such as O 1s and O 2s, respectively. The auger of Ga (Ga LMM) and oxygen (O KLL) were also observed in the figure. Figure 3b represents the atomic ratio of Ga 3d, O 1s, and C 1s for Ga_2_O_3_ films with the variation of substrate temperatures. From the figure, it can be seen that the percentage of O 1s element was nearly 50% and more than 30% for the Ga element. The C element was slightly high because there was no Ar etching before the measurement. The Ga/O ratio of all the films is about 0.7, which is close to the stoichiometric ratio of Ga_2_O_3_ and consistent with the report [28] of 0.71.

Ga 3d high-resolution XPS spectra taken from the surface of Ga_2_O_3_ films were fitted with two subpeaks, which are shown in Figure 3c. Ga 3d located at 20.54 eV is a typical value for Ga-O bonding in β-Ga_2_O_3_. The binding energies of the oxidation valence state of Ga^3+^ and Ga^+^ are 19.98~20.18 eV and 18.08–19.38 eV, respectively. Ga^3+^ corresponds to a high oxidation state in the film, while Ga^+^ relates to a low oxidation state [23], as shown in Figure 3c. The proportion of Ga^3+^ increases with the increase in substrate temperature and reaches the highest value of 92.42% at 250 °C. It indicates that the film quality gets better and better with the increase in substrate temperature. Figure 3d shows the core O1s energy level spectra of Ga_2_O_3_ films. The fitting is divided into two peaks. The binding energy at 530.7 eV conforms to the chemical bonding state of the Ga_2_O_3_ film, namely lattice oxygen (O_L_). Moreover, the other peak located at 531.3 eV [35] is due to oxygen deficiency (O_D_). Figure 3e,f shows the peak area ratio of Ga^3+^, Ga^+^, O_L_, and O_D_, respectively. When the substrate temperature rises from 80 to 250 °C, the proportion of O_L_ increases from 82.27% to 87.05%, and the proportion of O_D_ decreases from 17.73% to 12.95%. When the substrate temperature is low, oxygen does not have enough energy to fully react with tmga, resulting in a high ratio of oxygen deficiency [35], which is a signal from lattice oxygen in the vicinity of lower valence cation or in the vicinity of oxygen vacancy. The increase in substrate temperature is beneficial for reducing the defects in Ga_2_O_3_ film. The higher the substrate temperature, the fewer the defects in the films. Hao Liu et al. [35] grew β-Ga_2_O_3_ thin films by PLD. When the oxygen pressure was 0.01 Pa, the lattice oxygen content in the films was 87.06%, and the Ga^3+^ content was 92.53%, which were close to the results obtained in our study. The Ga_2_O_3_ film prepared using PLD by HongYang [36] et al. under the condition of 400 °C has 79.5% Ga-O content and ~20.5% C-O content. The Ga content of Ga_2_O_3_ is 91.9%. Yancheng Chen et al. [37] prepared Ga_2_O_3_ films on c-plane sapphire substrates using a plasma-enhanced chemical vapor deposition technique; they reported that the lowest oxygen deficiency ratio was 27.8%. Obviously, the Ga_2_O_3_ film prepared in this study possesses a low defect density, which is superior to the above preparation methods.

Figure 4 is the SEM images of the top view of Ga_2_O_3_ thin film. The images show that Ga_2_O_3_ films grown at different substrate temperatures are very smooth. Moreover, the AFM measurement was further used to investigate the roughness of the films. The AFM images were inset on the top right corner of the SEM images. Although the film is smooth and mostly in an amorphous state [34], it can still be found that the root mean square roughness (Rq) of the Ga_2_O_3_ thin film increases from 0.091 to 0.187 nm with the increase in substrate temperature [38]. Combined with XRR measurements, the Ga_2_O_3_ surface is relatively smooth. Compared with the roughness of the Ga_2_O_3_ film (about 0.8 nm) prepared by RF Sputtering [8], the smoother Ga_2_O_3_ film can be obtained in this study.

Figure 5 shows the cross-sectional TEM images of the PEALD Ga_2_O_3_ film prepared at 250 °C. In Figure 5a, a sandwich structure can be observed, which consists of three regions: the Ga_2_O_3_ layer, interfacial oxide SiO_2_, and Si substrate. According to the study by H. Altuntas et al. [39], a very thin SiO_2_ layer at the Si/Ga_2_O_3_ interface was formed. Many researchers have illustrated that Ga_2_O_3_ films deposited by PEALD at the window temperatures were amorphous. Inci Donmez et al. [40] pointed out that gallium oxide prepared by PEALD at 28–400 °C was amorphous when the film was deposited at 250 °C. Richard O’Donoghue et al. [34] deposited Ga_2_O_3_ at 60–160 °C by PEALD and also reached the same conclusion. Ranjinth k. Ramachandran et al. [20] deposited gallium oxide films at 100–400 ℃, and all the films were amorphous. Compared with X. Li et al.’s [28] investigation of the deposition of Ga_2_O_3_ by PEALD, it is interesting to find that even within the same substrate temperature variation range, slight crystallization can be seen at 250 °C in our study through high resolution-scanning/transmission electron microscopy (HR-S/TEM) measurements. Therefore, PEALD deposition is difficult to form on highly crystalline Ga_2_O_3_ films. A highly compact Ga_2_O_3_ thin film with a thickness of about 38 nm can be observed, which is in good agreement with the ellipsometer measurement result. The film is dense without any observable vacancies or voids. Figure 5b illustrates a magnified micrograph of Figure 5a. It can be clearly seen that the deposited Ga_2_O_3_ thin films have some nanocrystalline structure. The lattice spacing of about 1.66 Å evaluated from the HR-S/TEM image corresponds to the (116) plane of the α-Ga_2_O_3_ film [41,42], which is in agreement with the result from the XRD measurements. The interface of Ga_2_O_3_/Si shows an amorphous SiO_2_ layer, which results from the O_2_ plasma radicals reacting with the substrate in the first few cycles. The structure of the interfacial SiO_2_ layer was also found in our previous studies and literature using PEALD to prepare metal oxide films on Si wafers [22,39]. Cross-assigned EDX elemental mapping of carbon (C), oxygen (O), and gallium (Ga) is shown in Figure 5c. From the image, the C signal is hard to find within the Ga_2_O_3_ layer. The O and Ga elements were uniformly distributed in the film.

Figure 6a shows the transmittance spectra of Ga_2_O_3_ films at various temperatures on the sapphire substrate. The transmittance spectra were measured in the wavelength of 200 to 800 nm. All of the Ga_2_O_3_ films display an excellent optical transmittance with a high average transmittance of more than 90% in the visible light range. The absorption edge presents a redshift with the increase in substrate temperature, which is concerned with the reduced band gap of the film. The band gap of the film was determined by spectrophotometry, which is shown in Figure 6b. The transmission spectrum is transformed into (*αhν*)^2^~*hν*, *α* is calculated by the Beer–Lambert law: *α*(*λ*) = ln(1/*T*(*λ*))*/d*, where *T* is the penetration rate, and *d* is the thickness of the film. By extrapolating the linear region of (*αhν*)^2^~*hν* to the horizontal axis, the band gap of the film was obtained. Figure 6c shows the decreasing band gap from 5.04 to 4.76 eV with the increase in substrate temperature. Yancheng Chen et al. [37] reported that the band gap of the Ga_2_O_3_ film was between 4.4 eV and 5.1 eV. At lower substrate temperatures, the higher band gap value is attributed to the effect of excess O_2_ in the film or the presence of amorphous properties [43]. However, XPS results show that the ratio of Ga/O in the Ga_2_O_3_ film prepared in this experiment is slightly more than 0.7. The slight decrease in the band gap may be attributed to the high substrate temperature resulting in the formation of some nanocrystalline structures in the film.

Figure 6d shows the refractive index changes of the Ga_2_O_3_ films prepared at different substrate temperatures at different wavelengths. The refractive index curves of thin films were fitted by using the Tauc–Lorentz model. The obtained refractive index at 632.8 nm is 1.85~1.9, which is in good agreement with other reports using the same PEALD process [28,34,44]. The refractive index of amorphous Ga_2_O_3_ films increases gradually with the increase in substrate temperature due to the increasing density of Ga_2_O_3_ film. In the range of growth temperature, the higher the growth temperature is, the more favorable it is for the gaseous by-products to be desorbed and discharged from the surface so as to obtain a higher density film. XRR results also show that the film density increases with the increase in substrate temperature.

## 4. Conclusions

Ga_2_O_3_ thin films were deposited using tmga and oxygen plasma by PEALD in a substrate temperature range from 80 to 250 °C. The films present with an amorphous structure deposited at 80 to 200 °C, and they had a slight crystallization at 250 °C. XRR analysis of the film density found that the higher the temperature was, the denser the films were. According to XPS analysis, the proportion of Ga^3+^ and O_L_ increases with the increase in temperature, which means the reduced defects of the film. Moreover, films with higher density and better quality can be obtained with the increase in temperature. Additionally, the band gap of the Ga_2_O_3_ thin films decreases with the increase in temperature. SEM and AFM results show that changing the substrate temperature has little effect on the surface morphology, and the surface of Ga_2_O_3_ films is smooth. HR-S/TEM images show that Ga_2_O_3_ films have some nanocrystalline structures corresponding to the (116) plane of the α-Ga_2_O_3_ film.

## Figures and Tables

**Figure 1 nanomaterials-12-01510-f001:**
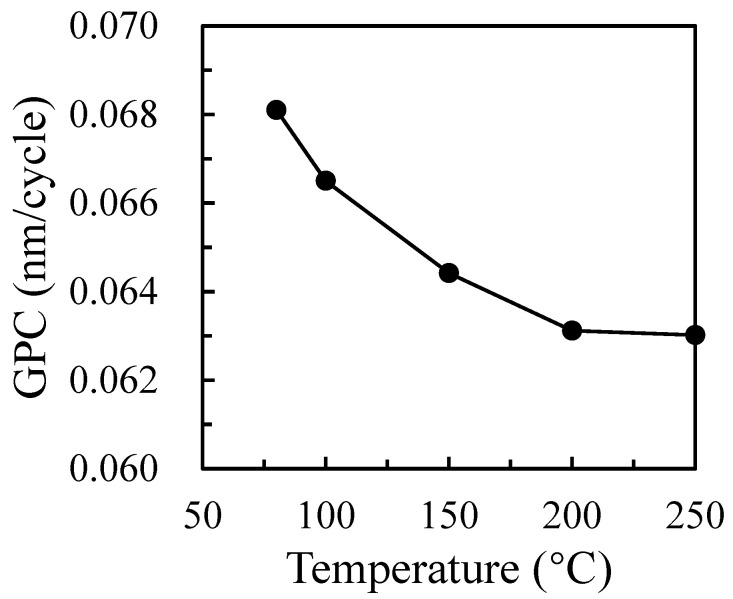
Growth rate of Ga_2_O_3_ thin films as a function of deposition temperature.

**Figure 2 nanomaterials-12-01510-f002:**
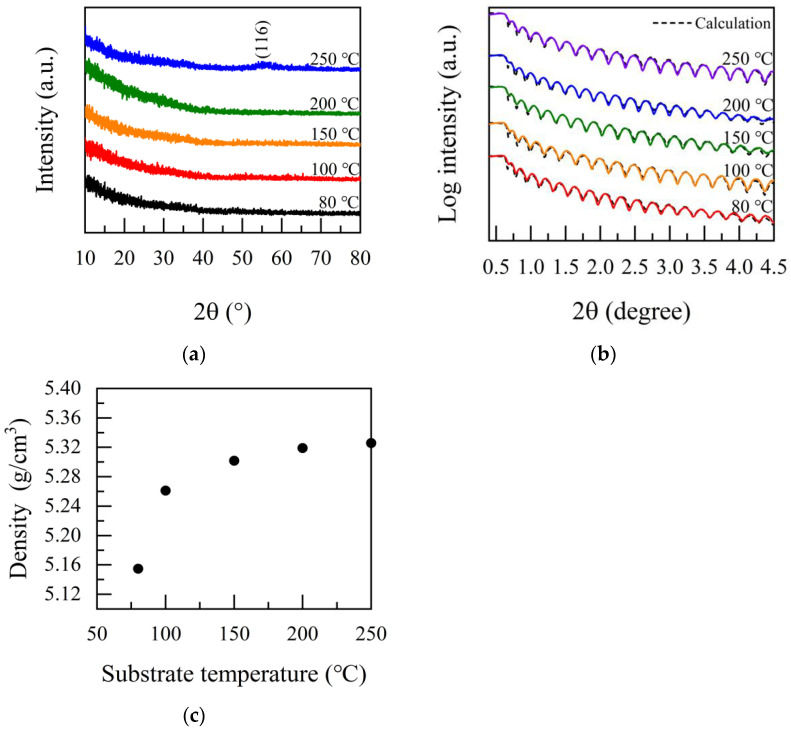
PEALD-Ga_2_O_3_ films grown at different substrate temperatures: (**a**) XRD measurements; (**b**) XRR measurements; (**c**) the films’ density.

**Figure 3 nanomaterials-12-01510-f003:**
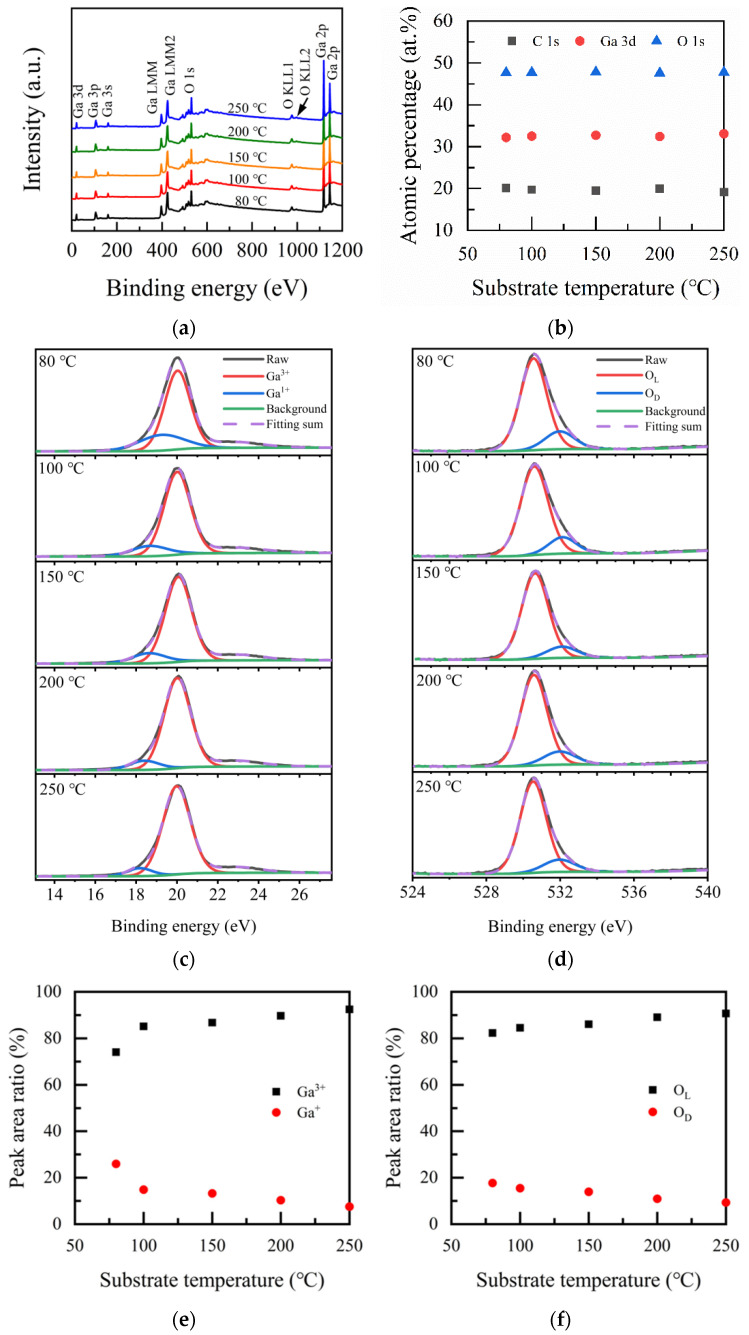
(**a**) XPS spectra of the Ga_2_O_3_ films deposited at different deposited temperatures. (**b**) Atomic ratio of the Ga_2_O_3_ films vs. substrate temperature. The high-resolution spectra of (**c**) Ga 3d; (**d**) O 1s; peak area ratios of (**e**) Ga^3+^ and Ga^+^ (**f**) O_L_ and O_D_.

**Figure 4 nanomaterials-12-01510-f004:**
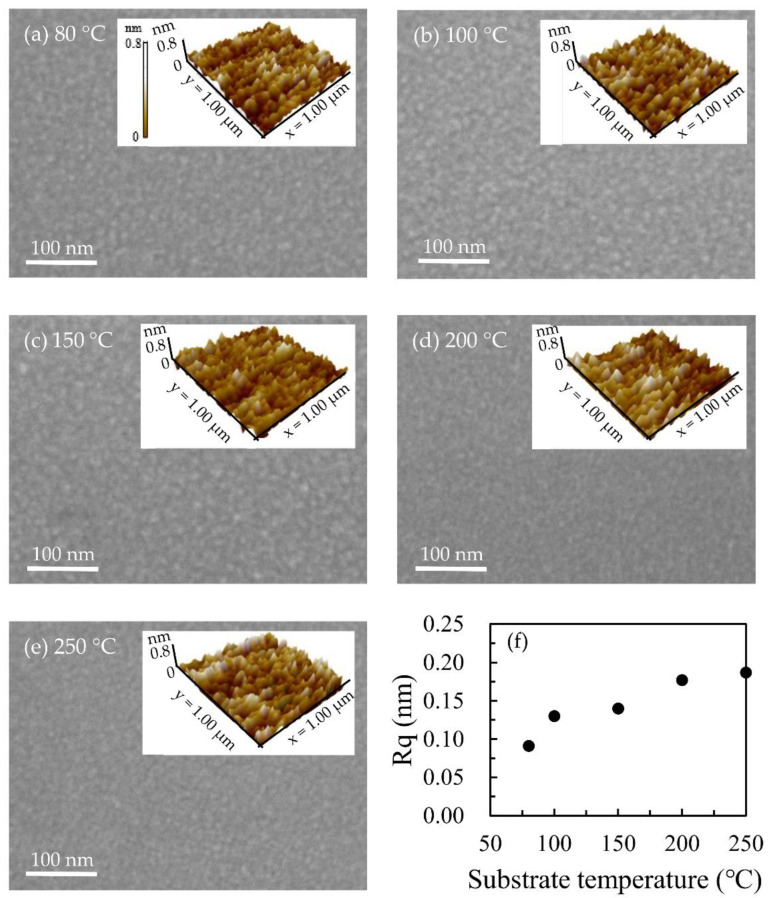
SEM and AFM images for (**a**) 80 °C, (**b**) 100 °C, (**c**) 150 °C, (**d**) 200 °C, and (**e**) 250 °C; (**f**) Rq of the Ga_2_O_3_ films as a function of the substrate temperature.

**Figure 5 nanomaterials-12-01510-f005:**
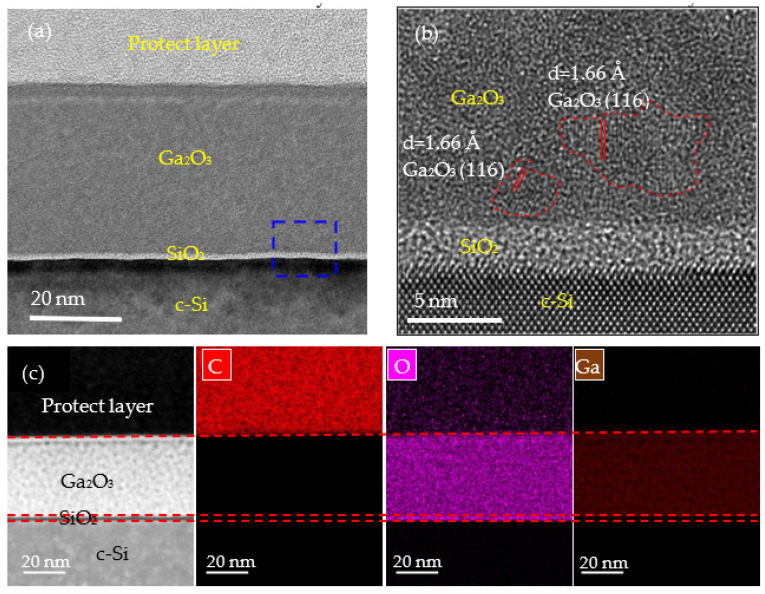
Cross-sectional HR-S/TEM images of a PEALD Ga_2_O_3_ film deposited at 250 °C at magnifications of (**a**) 245 k and (**b**) 1050 k. The areas marked by red lines indicate the nanocrystalline structure. (**c**) EDX elemental mapping for the as-deposited Ga_2_O_3_ film.

**Figure 6 nanomaterials-12-01510-f006:**
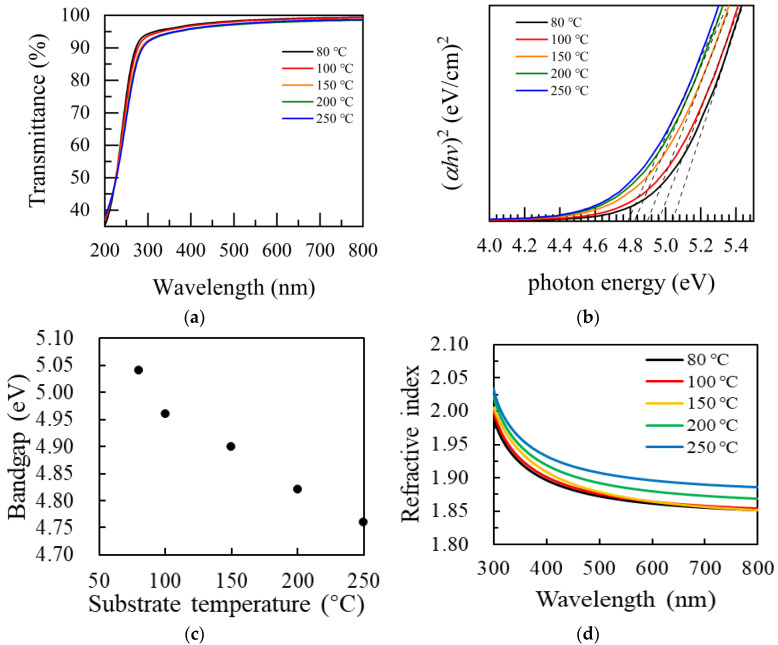
(**a**) Transmittance spectra of Ga_2_O_3_ prepared under various deposited temperatures. (**b**) Plots of (*αhν*)^2^ as a function of photon energy (*hν*); (**c**) Variation of the optical bandgap with the substrate temperature; (**d**) SE model results of the refractive index as a function of deposition temperature for Ga_2_O_3_ films.

**Table 1 nanomaterials-12-01510-t001:** Deposition conditions of PEALD Ga_2_O_3_ films.

Parameters	Value
Bubbler temperature (°C)	0
Substrate temperature (°C)	80–250
tmga pulse time (s)	0.2
tmga purge time (s)	4
O_2_ pulse time (s)	28
O_2_ flow stabilization (s)	1.4
O_2_ RF power on (s)	26
O_2_ purge time (s)	4
Flow rate of Ar (sccm)	80
Flow rate of O_2_ (sccm)	380
O_2_ plasma power (W)	2500
tmga carry gas (sccm)	120
tmga dilute gas (sccm)	400

**Table 2 nanomaterials-12-01510-t002:** Density and roughness of Ga_2_O_3_ films grown by diverse deposition techniques. Reported data of this work are included for comparison.

Deposition Technique	Thickness	Density (g/cm^3^)	Roughness (nm)	Temperature (°C)	Ref.
RF sputtering	89.9–103	5.28	0.88	500	[8]
MBE	12.7–95.4	5.30 ± 0.06	0.2–0.3	350–500	[31]
e-beam evap.	40	5.15	-	40	[32]
PEALD	30	5.3	0.44	120	[34]
PEALD	37.8	5.33	0.187	250	This work

## Data Availability

No new data were created or analyzed in this study. Data sharing is not applicable to this article.

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
