# Peer review of "Compact Ga_2_O_3_ Thin Films Deposited by Plasma Enhanced Atomic Layer Deposition at Low Temperature"

_nanomaterials, 2022, doi:10.3390/nano12091510_

Round 1

Reviewer 1 Report

Abstract and conclusions are based on the results and acceptable. Results are important for the gallium oxide synthesis and characteristics. Methodology is properly chosen and measurements well described.

Introduction – Aims of study should be formulated in  the text.

Ga(TMHD)3  tmhd  and TMGa -  I do prefer ligands in small case and  abbreviations of  tmhd and tmg must be placed in the text.

Figure 3 c and d “Raw´” line is not visible on the drawing.

tauc-lorentz - these are names should be Tauc – Lorentz.

Paper can be acceptable after minor corrections.

Author Response

Point 1: Introduction – Aims of study should be formulated in the text.

 Response 1: Thanks for your comments.

“The effects of different substrate temperatures on the deposition mechanism and properties of Ga2O3 thin films were discussed to obtain high density Ga2O3 thin films with few defects”. This description has been added in the Introduction section.

Please see Line 77-79 on Page 2.

Point 2: Ga(TMHD)3tmhd and TMGa - I do prefer ligands in small case and abbreviations of tmhd and tmg must be placed in the text.

Response 2: Thanks for your comments.

Gallium tetramethylheptanedionate (Ga(tmhd)3) and trimethylgallium (tmga) were revised and added in the manuscript.

Please see Line 69-70 on Page 2.

Point 3: Figure 3 c and d “Raw´” line is not visible on the drawing.

Response 3: Thanks for your comments.

Due to good Fitting condition, Raw and Fitting sum almost completely coincide. The Raw are solid black lines and Fitting data are dashed purple lines.

Please see the revised Fig 3(c) and (d).

Point 4: tauc-lorentz - these are names should be Tauc – Lorentz.

Response 4: Thanks for your comments.

“The refractive index of the Ga2O3 thin films were obtained from the SE characterization assuming a Tauc-Lorentz model.”

Please see Line 108 on Page 3.

Reviewer 2 Report

The paper Compact Ga2O3 Thin Films Deposited by Plasma Enhanced Atomic Layer Deposition at Low Temperature presents solid and soundly described investigation. I can't say that this is a particularly original or exciting research. However, it was carried out systematically and correctly and brings some shifts in the results reported so far. All conclusions seem to me sound and corroborated with experimental findings. As another good side of the manuscript, I can point out the abundant references in the discussion, which increase the quality of the discussion.

As a bad side, I can mention certain language clumsiness, e.g.: which resulting, was slightly crystal, tauc-lorentz, becomes to be intensified, or some localized microstructure, the bonding state, owns low defects, roughness increasing from, appear to be comparatively weak, wonderful optical transmittance etc. Therefore, the article should be proofread by a language expert.

I would also suggest an explanation for the attribution of the XPS peak to oxygen deficiency. Signal cannot be attributed to vacancy since vacancy cannot give XPS signal. I believe that is a signal from lattice oxygen in the vicinity of lower valence cation or in the vicinity of oxygen vacancy. A sentence clarifying this notion would be of use.

Also, the comparison of results obtained with literature reference in Introduction (Compared with X. Li…) is premature.

The title of Title 1 (Deposition contents…) should be changed, maybe Deposition conditions?

The title of Fig. 4 is somewhat baffling. SEM, AFM and roughness are given (and discussed in text). Not refractive index.

To conclude. I suggest publishing the reviewed paper in Nanomaterials after minor revision.

Author Response

Point 1: which resulting, was slightly crystal, tauc-lorentz, becomes to be intensified, or some localized microstructure, the bonding state, owns low defects, roughness increasing from, appear to be comparatively weak, wonderful optical transmittance etc. Therefore, the article should be proofread by a language expert.

 Response 1: Thanks for your comments.

All these descriptions have been revised and approved by a language expert.

Point 2: I would also suggest an explanation for the attribution of the XPS peak to oxygen deficiency. Signal cannot be attributed to vacancy since vacancy cannot give XPS signal. I believe that is a signal from lattice oxygen in the vicinity of lower valence cation or in the vicinity of oxygen vacancy. A sentence clarifying this notion would be of use.

Response 2: Thanks for your comments.

“When the substrate temperature is low, oxygen does not have enough energy to fully react with tmga, resulting in a high ratio of oxygen deficiency [36], which is a signal from lattice oxygen in the vicinity of lower valence cation or in the vicinity of oxygen vacancy.” This description has been revised in the manuscript.

Please see Line 224-227 on Page 6.

[36] H. Liu, C. Xu, X. Pan, and Z. Ye, Journal of Elec Materi 49, 4544 (2020).

Point 3: Also, the comparison of results obtained with literature reference in Introduction (Compared with X. Li…) is premature.

Response 3: Thanks for your comments.

“Compared with X. Li et al [21]. 's investigation on the deposition of Ga2O3 by PEALD, it’s interesting to find that even within the same substrate temperature variation range, slight crystallization can be seen at 250 ℃ in our study through high resolution-scanning/transmission electron microscopy (HR-S/TEM) measurement.” These descriptions have been added to “Results and discussion” section in the revised manuscript.

Please see Line 263-266 on Page 9.

[21] X. Li, H.-L. Lu, H.-P. Ma, J.-G. Yang, J.-X. Chen, W. Huang, Q. Guo, J.-J. Feng, and D.W. Zhang, Current Applied Physics 19, 72 (2019).

Point 4: The title of Title 1 (Deposition contents…) should be changed, maybe Deposition conditions?

Response 4: Thanks for your comments.

Table 1. Deposition conditions of PEALD Ga2O3 films.” The description has been added to the revised manuscript.

Please see Line 104 on Page 3.

Point 5: The title of Fig. 4 is somewhat baffling. SEM, AFM and roughness are given (and discussed in text). Not refractive index.

Response 5: Thanks for your comments.

Figure 4. SEM and AFM images for (a) 80 oC, (b) 100 oC, (c) 150 oC, (d) 200 oC and (e) 250 oC; (f) Rq of the Ga2O3 films as a function of the substrate temperature.” The original descriptions of Figure 4 have been changed to the above descriptions.

Please see Line 252-253 on Page 8.
